# Adaptation and Validation of the Diabetic Foot Ulcer Scale-Short Form Scale for Chinese Diabetic Foot Ulcers Individuals

**DOI:** 10.3390/ijerph192114568

**Published:** 2022-11-06

**Authors:** Lin Ma, Wanxia Ma, Shuang Lin, Yan Li, Xingwu Ran

**Affiliations:** 1West China School of Nursing, Sichuan University, Chengdu 610041, China; 2Department of Endocrinology and Metabolism, West China Hospital, Sichuan University, Chengdu 610041, China; 3Chengdu Second People’s Hospital, Chengdu 610017, China; 4Innovation Center for Wound Repair, Diabetic Foot Care Center, Department of Endocrinology and Metabolism, West China Hospital, Sichuan University, Chengdu 610041, China

**Keywords:** DFS-SF, adaptation, validity, reliability, Chinese, quality of life

## Abstract

Background: The quality of life (QoL) of diabetic foot ulcer patients is worse than that of diabetic patients. The Diabetic Foot Ulcer Scale-Short Form (DFS-SF) is a readily available instrument used to evaluate the quality of life of diabetic foot ulcer individuals. The aim of this study was to translate the DFS-SF into Chinese, followed by an evaluation of its validity and reliability. Methods: This study was conducted in two phases. In the first phase, we followed the Brislin’s Translation and Back-translation model to translate the DFS-SF into Mandarin Chinese. In the second phase, we examined the reliability and validity of the Chinese version of the DFS-SF, where the reliability was assessed in terms of Cronbach’s α coefficient, split-half reliability, and test-retest reliability, and validation of the scale was carried out through content validity, structure validity and criterion validity approaches. Results: A total of 208 participants were recruited for our study. The item-level content validity index (I-CVI) of the Chinese version of the Diabetic Foot Ulcer Scale varied from 0.800 to 1.000, the average scale-level content validity index (S-CVI/Ave) was 0.911, and the Cronbach’s α coefficient of the scale was 0.952. Confirmatory factor analysis indicated good structural validity of the scale, with a Comparative Fit Index (CFI) = 0.920 and a root mean square error of approximation (RMSEA) of 0.069 (*p* < 0.001). The criterion-related validity results indicated that the subscales were significantly related to the subscales of the 36-Item Short-Form Health Survey (SF-36), with coefficients ranging from 0.116 to 0.571 (*p* < 0.05). Discussion: The translation and the examination of the scale rigidly followed the golden standard model, and the reliability observed in our study was similar to that of other studies. Furthermore, the validity assessment indicated that the scale structure was reliable. Therefore, the proposed scales may serve as a reliable instrument for the quality of life evaluation in the diabetic foot ulcers population. Conclusion: The adaptation and validation of the Chinese version of the Diabetic Foot Ulcers Scale-Short Form were reliable, and it will be a reliable instrument to evaluate the QoL of Chinese diabetic foot ulcer patients.

## 1. Introduction

Diabetic foot ulcers (DFUs), which are a common and severe complication of diabetes mellitus, have been associated with increased risk of morbidity and mortality [1,2]. They also impose a physical, mental, and economic burden on the affected population, and diabetic foot ulcer individuals suffer greatly in their daily life. Therefore, a better assessment of their quality of life is needed, in order to provide further support to them.

The quality of life of diabetic foot ulcer patients is affected by their multi-disciplinary situation and treatment. Many studies have reported that the quality of life for people with diabetic foot ulcers is worse than that of diabetic people and the general population in terms of physical and psychological factors [3,4], and the measurements of quality of life vary. Nevertheless, there are few instruments focused on the quality of life of diabetic foot ulcer patients.

The Diabetic Foot Ulcer Scale-Short Form (DFS-SF) was developed from the Diabetic Foot Ulcer Scale (DFS). The DFS [5] contains 64 items to evaluate the quality of life of those affected by diabetic foot ulcers. Dr. Bann has promoted the final DFS-SF as a more brief and convenient alternative [6]. It consists of 29 items with six sub-scales: leisure, physical health, dependence/daily life, negative emotions, worried about ulcer/feet, and bothered by ulcer care. Each item is scored followed a 5-point Likert scale from 1 to 5 points, ranging from “not at all” to “very common”, respectively. A higher score indicates a higher quality of life among diabetic foot ulcers individuals. The scale has been translated and tested for reliability and validity in Spain [7], Greece [8], Turkey [9], Korea [10], and the Netherlands [11]. However, in China, there is no specific validated instrument to assess the QoL of patients with DFUs at present. Therefore, for this study, we aimed to translate the Diabetic Foot Ulcer Scale-Short Form into Mandarin Chinese and test its validity among Chinese diabetic foot ulcer individuals.

## 2. Methods

### 2.1. Design

The study was carried out in two steps. Our first aim was to produce a Chinese version of the Diabetic Foot Ulcer Scale-Short Form (DFS-SF), based on an established translation and cross-cultural considerations. Our second purpose was to examine the reliability and validity of the Chinese Revised DFS-SF scale, and apply it to Chinese diabetic foot ulcer individuals with well-designed measurements.

### 2.2. Quality of Life Measurements

#### 2.2.1. Diabetic Foot Ulcer Scale-Short Form

After connecting with the Mapi Research Trust, we obtained the Diabetic Foot Ulcer Scale-Short Form in English. Then, we translated it into Mandarin Chinese using Brislin’s model [12] of forward and backward translation by five translators: First, a nursing graduate who did well in English and an English language professor with no medical background translated the scale into Chinese, forming the primary Chinese version of the scale. Then, after an expert in diabetic foot ulcers evaluated the content equivalence, we produced the second Chinese version of the DFS-SF scale. To ensure the semantic equivalence of the Mandarin Chinese translation, a doctor unfamiliar with the original questionnaire with a bilingual background and an English-Chinese translation expert independently translated the second Chinese version back to an English version. After these two translators reached an agreement, the product was the back-translated from the English version.

After the back-translation, these five experts examined the meanings of the items among the original Diabetic Foot Ulcer Scale-Short Form, the back-translated Diabetic Foot Ulcer Scale-Short Form, and the second Chinese translation of the Diabetic Foot Ulcer Scale-Short Form (DFS-SF-C). The revised version was then used to pre-test 30 diabetic foot ulcer individuals, in order to confirm that the items were easy to understand and met Chinese language habits. The participants expressed that the items on the scale were easy to understand. After some minor adjustments and modifications, the final Chinese version of the Diabetic Foot Ulcer Scale-Short Form (DFS-SF-C) was adopted for the current study.

#### 2.2.2. 36-Item Short-Form Health Survey (SF-36)

The SF-36 contains 36 items [13] and is a standard scale used to measure the quality of life among the general population in any situation. It consists of nine sub-scales: Physical Functioning (PF), Role-Physical (RP), Bodily Pain (BP), General Health (GH), Validity (VT), Social Functioning (SF), Role-Emotional (RE), Mental Health (MH), and Reported Health Transition (HT).

### 2.3. Psychometric Testing

#### 2.3.1. Participants

Confirmatory Factor Analysis (CFA) [14] was conducted to test the internal consistency [15]. As the CFA is performed based on structural equation modelling (SEM) [16], sampling followed the SEM sampling calculation method, where the sample size should be at least 200 [17]. Also, it was suggested that the sample of the CFA should be between 100 and 200 [10]. Based on all of these factors, we decided that the sampling size should be at least 200. In this respect, the sample size in our study was considered efficient.

We performed a multi-center cross-sectional study. Through a convenient sampling approach, we selected patients recruited from the West China Hospital of Sichuan University, the Sichuan Provincial People’s Hospital, the Fourth People’s Hospital of Sichuan Province, the Chengdu First People’s Hospital, and Chengdu Second People’s Hospital from April 2021 to February 2022. All participants met the following inclusion criteria: (1) diagnosed with diabetic foot ulcers; (2) age ≥ 18; (3) conscious and could be appropriately communicated with; and (4) provided informed consent and volunteered to participate in the research. The exclusion criteria were as follows: (1) hearing impairment; (2) cognitive impairment; (3) inability to cooperate with the completion of this study; and (4) serious post-operative complications.

#### 2.3.2. Validity

Ten experts then reviewed the content validity of the translated DFS-SF. They evaluated the relevance of each item to the corresponding sub-scale. The content validity was established on a 4-point rating scale (1 = not relevant, 2 = somewhat relevant, 3 = relevant, and 4 = significantly relevant). The item-level content validity index (I-CVI) is the proportion of total items evaluated by the experts as either 3 or 4, with a value of greater than 0.8 indicating good content validity. The average scale-level content validity index (S-CVI/Ave) was also used to describe the content validity. Confirmatory factor analysis (CFA) was selected to evaluate the construct validity. An $x^{2}$*/df* value of <3.000, comparative fit index (CFI) of >0.900, and root mean square error of approximation (RMSEA) of <0.080 indicate good model fit [18,19]. The criterion-related validity test was conducted with reference to the SF-36.

#### 2.3.3. Reliability

Cronbach’s alpha coefficient [20], split-half reliability, and the test-retest reliability were used to describe the internal consistency of the total scale and six sub-scales. The test-retest reliability was conducted among the 30 pre-test participants. After the first test, we tested the scale among them again. Values in the range of 0.70–0.80 are considered acceptable, but a value over 0.90 was to be expected [21,22].

#### 2.3.4. Correlation

The product–moment correlation method was used to measure the correlation among each sub-scale of the DFS-SF scale. Pearson correlation analysis was conducted to evaluate criterion validity. The correlation was valued as follows: weak correlation (0.10–0.29), moderate correlation (0.30–0.49), and strong correlation (0.50–1.00) [23].

### 2.4. Data Analysis

Quantitative data were analyzed using SPSS V.23 and Amos 23.0 (IBM Corp, Armonk, New York, NY, USA). Continuous variables were given as means and standard deviations, while categorical variables were reported as frequencies and percentages. A value of *p* < 0.05 was considered to indicate statistical significance.

### 2.5. Ethical Considerations

This study received ethical approval from the Ethics Committee on Biomedical Research of the West China Hospital, in Sichuan University. All the participants in this study readily volunteered, and their private information was anonymized. They all signed an informed consent form after being informed of the significance of the study.

## 3. Results

### 3.1. Characteristics of the Participants

We recruited a total of 246 patients, 23 of whom rejected inclusion, for a response rate of 90.7%, while 15 participants did not finish the entire assessment. Finally, 208 patients who finished the assessment were included in our study. The average scores on the DFS-SF scale were 98.88 ± 25.904. The mean age of the participants was 64.37 (ranging from 23–93); a total of 30.3% (63) were female, and 69.7% (145) were male. The details of the participants’ characteristics are given in Table 1.

### 3.2. Validity

#### 3.2.1. Content Validity

Ten experts assessed the content validity. The I-CVI should be valued from 0.800 to 1.000; however, there were three items that tested below 0.800. After communicating further with experts, we decided to delete the item “Drained” (which I-CVI scored 0.500). As most experts suggested, this item repeated the item “Fatigued or Tired”. Therefore, the final Chinese version of the Diabetic Foot Ulcer Scale-short Form consisted of 28 items, which is less than in the original scale. The construct validity and reliability were determined based on the revised Chinese scale. The final S-CVI/Ave was 0.911.

#### 3.2.2. Structure Validity

We performed confirmatory factor analysis (CFA) to confirm the structural validity. The model is shown in Figure 1.

The data fit the model well, where $x^{2}$*/df* = 1.995 (*p* < 0.001), RMSEA = 0.069 (a value less than 0.080 indicates a good model fit), and CFI = 0.920 (a value greater than 0.900 is indicative of a good model fit).

#### 3.2.3. Criterion-Related Validity

We conducted Pearson’s correlation analysis with the SF-36 scale, in order to test the criterion-related validity. The total scores of the DFS-SF were highly correlated with the total scores of the SF-36, with a coefficient of 0.586 (*p* < 0.001). The correlation coefficients for the sub-scales of the DFS-SF and SF-36 are provided in Table 2.

### 3.3. Reliability

Table 3 shows the reliability of the DFS-SF. The reliability test included the Cronbach’s α coefficient, split-half reliability, and test-retest reliability. A reliability value greater that 0.70 would be acceptable. The full-scale Cronbach’s α coefficient was 0.952, and the sub-scale Cronbach’s α coefficients ranged from 0.772 to 0.923.

### 3.4. Correlation

The correlation of each sub-scale is shown in Table 4. The correlation between the six sub-scales was significantly increased, from 0.346 to 0.758 (*p* < 0.001).

## 4. Discussion

The findings of our study indicate that the Chinese revision of the Diabetic Foot Ulcer Scale-Short Form is a suitable measurement tool to assess the quality of life in diabetic foot ulcers individuals. The translation process strictly followed the international scale introduction procedure [24,25]—namely, the methods of “translation– back-translation–cultural debugging–pretest”, which formed the Chinese version of the DFS-SF (DFS-SF-C). After conducting the content validity test, we deleted an item from the scale. We formed the final Chinese revision scale (DFS-SF-CV).

The sample size of our study was moderate, and we carried out a considerable number of tests to assess the validity and reliability. It has been reported that 110 participants were recruited in the Greek translation, and their validation test only included criterion-related structure validation [8]. There were 194 participants in the Turkish study, and content validity and Exploratory Factor Analysis (EFA) were conducted. Their EFA test confirmed that the original structure of DFS-SF with six sub-scales was maintained [9]. A total of 143 DFU patients were included in the Spanish study, and the validation was conducted with respect to Pearson’s correlation analysis with the SF-36 and EQ-5D, as well as construct validity testing through CFA and EFA, with acceptable results [7]. A relative study in Brazil recruited 290 participants [26], while there were 320 participants in a Korean study, 131 of which were healed DFU patients [10]. These studies conducted validation through content validity and construct validity (EFA and CFA) analyses.

In our tests, three items presented an I-CVI below 0.800; furthermore, the I-ICV scores of the items “meant that you had to spend more time planning and organizing for leisure activities” and “drained” were 0.500, while the I-ICV score of the item “frustrated by others doing things for you when you would rather do things yourself” was 0.700. In the Turkish study, they considered a correlation over 0.30 to be indicative of not requiring revision [9]. After consulting with the DFU experts, we decided to only exclude one item, “drained”, which repeated the meaning of the item “fatigue and tired”, regardless of any word we tried to translate this item into Chinese. Nevertheless, we retained the other two items, as we considered them significant to their sub-scales. We also compared the final results with or without the excluded items. There was not a large distance between the two results. Compared with other studies, the CVI in this study was a little bit lower, but still acceptable. The CVI for the Turkish version was a little higher than that of our study, at 0.97 [9], while the CVI for the Korean version was 0.98 [10].

As the DFS-SF is a mature scale, we did not conduct the exploratory factor analysis (EFA) to test the validity of the scale, and instead only chose confirmatory factor analysis (CFA) to confirm the structure of the scale. The analysis result was in agreement with the other relevant studies [6]. The CFI and RMSEA were reported as 0.844 and 0.095 for the Spanish version [7], respectively, which were similar to those in our study. The Korean study reported GFI, CFI, and RMSEA values of 0.73, 0.92, and 0.10, respectively. Overall, the structural validity of the DFS-SF scale was reliable.

The criterion validity should choose a gold standard scale for evaluating the quality of life. In the study of Dutch and Spanish studies [7,11,27], the EQ-5D scale was chosen for this purpose, with correlation values reported in the ranged of 0.166–0.454 [7], while a Jordanian study compared the SF-8 scale [28]. We decided to follow the other studies [6,7,8,10,29], in choosing the SF-36 as the gold standard. The correlation between the proposed DFS-SF scale and the SF-36 scale was weak or moderate, with coefficients ranging from 0.154–0.571 *(p* < 0.001); these results are not optimal [30,31]. For the results reported for the Greek version, the correlation coefficients between the DFS-SF and SF-36 ranged from 0.39–0.79, with most correlations being strong (i.e., above 0.70) [8]. In other studies, the correlations between the DFS-SF and SF-36 were moderate, ranging from 0.178–0.737 [7], 0.127–0.641 [26], or 0.24 to 0.54 [10]. All of these results were similar to our studies. Furthermore, our results were similar to those obtained in Bann’s study, which ranged from 0.24 to 0.62 [6]. The DFS-SF was moderately related to the SF-36. The reason for this may be that the SF-36 is more prevalent in the general population, and lacks NICE approval for use in utility studies [32]. Therefore, the DFS-SF is more acceptable for DFU people, with 29 items and higher sensitivity.

The proposed Chinese version of the DFS-SF was found to be reliable, and we can conclude that the internal consistency was acceptable. The Cronbach’s α in our study ranged from 0.772 to 0.952, similar to that in other studies: it has been reported in the range of 0.79–0.94 (Greek) [8], 0.83–0.94 (Turkish) [9], 0.720–0.948 (Spanish) [7], 0.72–0.89 (Brazil) [26], 0.96–0.99 (Korean) [10], and 0.82 to 0.93 (Polish) [33].

After translating and testing the scale, we kept the scale’s original structure and deleted one item. The validity of the Diabetic Foot Ulcer Scale-Short Form’s structure was acceptable. Moreover, it was found to be suitably related to the SF-36. Therefore, it can be considered a reliable instrument to measure the quality of life in the Chinese DFU population.

There was inevitable bias in this study. In the process of the study, we carried out strict quality control. Firstly, all of the participants in our study volunteered and cooperated with us, the inclusion of participants was completely randomized, and we rigidly followed the inclusion criteria. Secondly, there were a total of three researchers to collect the data, all of whom were taught the norm of the DFS-SF scale, and the data collection process was normalized and consistent. Thirdly, we did the re-test among 30 patients to test the reliability and to avoid potential bias.

It should be noted that there were some limitations in our study; for example, the number of samples was only just sufficient for the data analysis. In further research, we intend to collect a large amount of data from DFU patients to obtain more comprehensive results.

## 5. Conclusions

The Diabetic Foot Ulcer Scale-Short Form is a reliable instrument to assess the quality of life in DFU patients. The revised Chinese version of this scale was considered to be more acceptable for the Chinese population. It can be used more widely in the future, in order to develop interventions to improve the quality of life of Chinese people with DFUs, and thus better support them.

## Figures and Tables

**Figure 1 ijerph-19-14568-f001:**
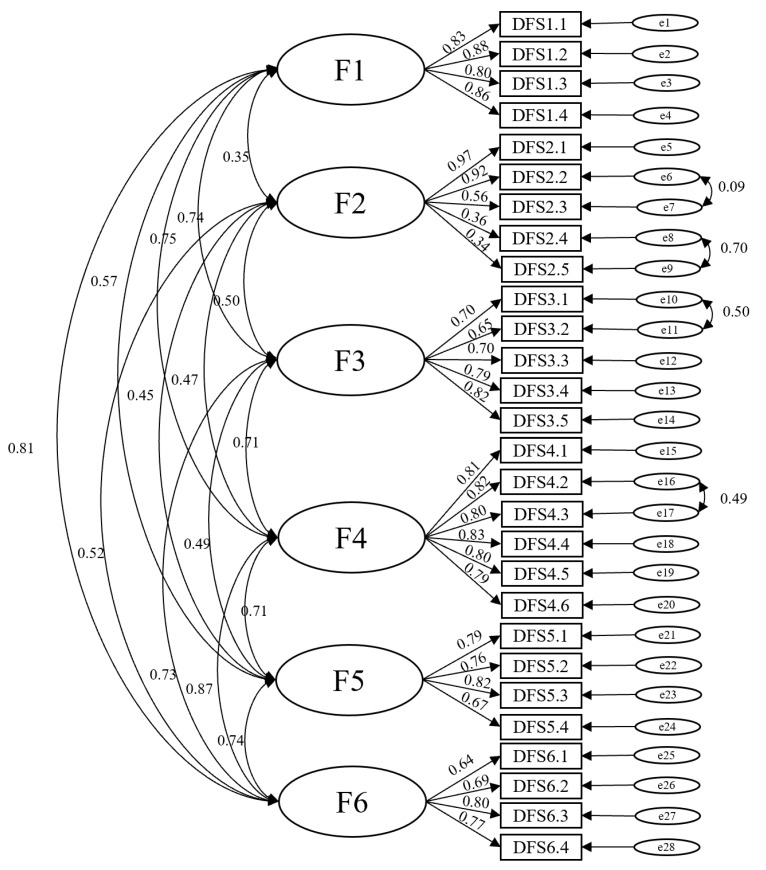
Confirmatory factor analysis of the different subscales of the Diabetic Foot Ulcer Scale-short Form (DFS-SF). (F1: Leisure; F2: Physical health; F3: Dependence/Daily life; F4: Negative emotions; F5: Worried about ulcers/feet; F6: Bothered by ulcer care.).

**Table 1 ijerph-19-14568-t001:** Characteristics of participants (N = 208).

Demographics		
Age		
Mean (SD)	64.4	13.1
Gender		
Male (n,%)	145	69.7
Female (n,%)	63	30.3
Education level		
Primary school or below (n,%)	87	41.8
Middle school (n,%)	62	29.8
High school (n,%)	31	14.9
College above (n,%)	28	13.5
Marriage		
Married (n,%)	172	82.7
Divorced (n,%)	4	1.9
Widows (n,%)	24	11.5
Unmarried (n,%)	8	3.9
Duration of diabetes		
<5 years (n,%)	31	14.9
5–10 years (n,%)	55	26.4
>10 years (n,%)	122	58.7
Whether amputation		
Yes (n,%)	40	19.2
No (n,%)	168	80.8
Duration of diabetic foot		
≤1 month (n,%)	59	28.4
1–≤3 months (n,%)	62	29.8
>3 months (n,%)	87	41.8
The number of ulcers		
1 (n,%)	96	46.2
>1 (n,%)	112	53.8
Whether the first ulcer occurrence		
Yes (n,%)	150	72.1
No (n,%)	58	27.9
Ulcer type		
Neuropathic (n,%)	77	37.0
Ischemic (n,%)	52	25.0
Neuro-ischemic (n,%)	79	38.0
Other diabetes complications		
Diabetic nephropathy (n,%)	37	17.8
Diabetic retinopathy (n,%)	85	40.9
Diabetic peripheral neuropathy (n,%)	152	73.1
Diabetic peripheral vascular disease (n,%)	81	38.9
Diabetic autonomic neuropathy (n,%)	93	44.7

**Table 2 ijerph-19-14568-t002:** Criterion-related validity of the Diabetic Foot Ulcer Scale-Short Form by Pearson’s correlation with the SF-36.

	HT	PF	RP	RE	SF	BP	VT	MH	GH
Leisure	0.079	0.030	0.136	0.173 **	0.278 **	0.234 **	0.242 **	0.288 **	0.275 **
Physical health	0.111	0.102	0.263 **	0.241 **	0.347 **	0.412 **	0.571 **	0.545 **	0.352 **
Dependence/daily life	0.133	0.227 **	0.181 **	0.222 **	0.307 **	0.278 **	0.376 **	0.387 **	0.416 **
Negative emotions	0.092	0.024	0.153 **	0.273 **	0.425 **	0.249 **	0.424 **	0.478 **	0.445 **
Worried about ulcers/feet	0.116 *	0.023	0.234 **	0.235 **	0.392 **	0.286 **	0.336 **	0.466 **	0.365 **
Bothered by ulcer care	0.118	0.125	0.194 **	0.246 **	0.389 **	0.228 **	0.372 **	0.452 **	0.345 **

* *p* < 0.005, ** *p* < 0.001. Abbreviations: PF, Physical Functioning; RP, Role-Physical; BP, Bodily Pain; GH, General Health; VT, Vitality; SF, Social Functioning; RE, Role-Emotional; MH, Mental Health; HT, Reported Health Transition.

**Table 3 ijerph-19-14568-t003:** Reliability of the DFS-SF.

	N	Cronbach’s α Coefficient	Spilt-Half Reliability	Test-Retest Reliability
Total	28	0.952	0.866	0.937
Leisure	5	0.923	0.896	0.888
Physical health	4	0.772	0.854	0.740
Dependence/daily life	5	0.870	0.768	0.710
Negative emotions	6	0.921	0.896	0.802
Worried about ulcers/feet	4	0.846	0.818	0.819
Bothered by ulcer care	4	0.815	0.803	0.811

**Table 4 ijerph-19-14568-t004:** Correlation of each subscale of DFS-SF.

	Leisure	Physical Health	Dependence/Daily Life	Negative Emotions	Worried about Ulcers/Feet	Bothered by Ulcer Care
Leisure	1.000					
Physical health	0.346 **	1.000				
Dependence/daily life	0.642 **	0.447 **	1.000			
Negative emotions	0.703 **	0.437 **	0.597 **	1.000		
Worried about ulcers/feet	0.490 **	0.446 **	0.403 **	0.627 **	1.000	
Bothered by ulcer care	0.706 **	0.469 **	0.589 **	0.758 **	0.616 **	1.000

** *p* < 0.001.

## Data Availability

The datasets used and/or analyzed during the current study are available from the corresponding author upon reasonable request.

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
