# Peer review of "Adaptation and Validation of the Diabetic Foot Ulcer Scale-Short Form Scale for Chinese Diabetic Foot Ulcers Individuals"

_ijerph, 2022, doi:10.3390/ijerph192114568_

Round 1

Reviewer 1 Report

This article addresses an important topic, especially given the burden of diabetic foot ulcers on diabetes management and mortality nowadays.

The title describes the topic well, while the abstract is well structured providing adequate information on the purpose of the study. The introduction explains the scientific background and gives us the specific objectives of the study. I would suggest using smaller phrases - they are easier to follow. Also, I consider it will be useful to describe more about validity tools and reliability tools.

Speaking of Materials and Methods, I would suggest you rearrange the values in Figure 1 (they are a little misplaced).

Discussions: What do you want to say in “meat that you had to spend more time planning and organizing for leisure activities”? I think the details should be clarified to ensure the readers understand exactly what the authors said.

I am interested in knowing your opinion about how you addressed sources of potential bias.

To sum up, this article would make an important contribution to scientific literature, validating a Scale for Diabetic Foot Ulcers for Chinese patients with diabetes mellitus. I consider you appropriately researched relevant sources and internationally validated scales. 

Author Response

Dear reviewer:

Thank you very much for your time and attention involved in the manuscript and your encouraging comments on the merits.

we also appreciate your clear and detailed feedback and hope that the explanation has adequately addressed all of your concerns. In the remainder of this letter, we will discuss each of your comments individually along with our corresponding responses.

To facilitate this discussion, we will first retype your comments in italics front and then present our responses to those comments.

Point 1: I would suggest using smaller phrases - they are easier to follow. Also, I consider it will be useful to describe more about validity tools and reliability tools.

Response:

Thanks for your detailed suggestion. We have revised the abstract as per your suggestion. Also, we added the description of validity and reliability tools in the methods part. The details are as follows:

 “Background: The quality of life (QoL) of diabetic foot ulcer patients is worse than that of diabetic patients. The Diabetic Foot Ulcer Scale-Short Form (DFS-SF) is a readily available instrument used to evaluate the quality of life of diabetic foot ulcers individuals. The aim of this study was to translate the DFS-SF into Chinese, followed by an evaluation of its validity and reliability.

Methods: This study was conducted in two phases: In the first phase, we followed the Brislin’s Translation and Back-translation model to translate the DFS-SF into Mandarin Chinese. In the second phase, we examined the reliability and validity of the Chinese version of the DFS-SF, where the reliability was assessed in terms of Cronbach’s α coefficient, Split-half reliability and test-retest reliability, and validation of the scale was carried out through content validity, structure validity and criterion validity approaches.

Results: A total of 208 participants were recruited for our study. The item-level content validity index (I-CVI) of the Chinese version of the Diabetic Foot Ulcer Scale varied from 0.800 to 1.000, the average scale-level content validity index (S-CVI/Ave) was 0.911, and the Cronbach’s α coefficient of the scale was 0.952. Confirmatory factor analysis indicated the good structural validity of the scale, with a Comparative Fit Index (CFI) =0.920 and root mean square error of approximation (RMSEA) of 0.069 (P <0.001). The criterion-related validity results indicated that the subscales were significantly related to the subscales of the 36-Item Short-Form Health Survey (SF-36), with coefficients ranging from 0.116 to 0.571 (P <0.05).

Discussion: The translation and the examination of the scale rigidly followed the golden standard model, and the reliability observed in our study was similar to that in other studies. Furthermore, the validity assessment indicated that the scale structure is reliable. Therefore, the proposed scales may serve as a reliable instrument for the quality of life evaluation in the diabetic foot ulcers population.

Conclusion: The adaptation and validation of the Chinese version of the Diabetic Foot Ulcers Scale-Short Form are reliable and it will be a reliable instrument to evaluate the QoL of Chinese diabetic foot ulcer patients.” (Page 1, Line 13-38)

Point 2: Speaking of Materials and Methods, I would suggest you rearrange the values in Figure 1 (they are a little misplaced).

Response:

Thanks for your detailed review. The details of the figure have been rearranged, and we have made improvements in this part: a more relevant line between e8 and e9 is made and described using standardized estimates. Figure 1 has been rearranged in detail, and the final improvement is as follows: (the Figure 1 please see in the the word)

Figure 1. Confirmatory factor analysis of the different subscales of the Diabetic Foot Ulcer Scale-short Form (DFS-SF). (F1: Leisure; F2: Physical health; F3: Dependence/Daily life; F4: Negative emotions; F5: Worried about ulcers/feet; F6: Bothered by ulcer care.)

The data of the model fit well. The /df was 1.995 (P<0.001), RMSEA=0.069 (stands for the root mean square error of approximation, and a value<0.080 indicates good model fit), CFI=0.920 (stands for comparative fit index, and a value>0.900 is indicative of good fit). (Page 7, Line 174-180)

Point 3: Discussions: What do you want to say in “meat that you had to spend more time planning and organizing for leisure activities”? I think the details should be clarified to ensure the readers understand exactly what the authors said.

Response:

Thanks for your detailed review. We feel so sorry for the misspelling of this sentence. This is an original item on the scale. The concept expression of this sentence is “meant that you had to spend more time planing and organizing for leisure activities” The meaning of this sentence is as follows: Because of the limitations of movement, it difficult for diabetic foot ulcer patients to get out independently, they need to spend time communicating with their caregivers and plan the time-consuming, plan transportation and route in their leisure time. (Page 9, Line 223-224)

Point 4: I am interested in knowing your opinion about how you addressed sources of potential bias.

Response:

Thanks for your detailed review. We feel so sorry for not clarifying this part cleraly, perhaps because we are not experts in data analysis. Thank you for giving us the opportunity to explain this. The patients included in this study were truly randomized and covered all statuses of diabetic foot ulcers to ensure the variety of samples. Moreover, in further study, we wil also collect more patients in our study and concduct more in-depth research.

Among the limitations of our study, we have revised this part as follows:

“It should be noted that there were some limitations in our study; for example, the number of samples was only just sufficient for the data analysis. In further research, we intend to collect a large amount of data from DFUs patients to obtain more comprehensive results.” (Page 10, Line 275-278)

We also added this prat in our discussion to reply this problem:

“There was inevitable bias in this study. In the process of the study, we carried out strict quality control. Firstly, all of the participants in our study volunteered and cooperated with us, as well as the inclusion of participants was completely randomized and rigidly followed the inclusion criteria rigidly. Secondly, there were a total of three researchers to collect the data, all of them were taught the norm of the DFS-SF scale, and the data collection process was normalized and consistent. Thirdly, we did the re-test among 30 patients to test the reliability to avoid potential bias.” (Page 10, Line 268-274)

Sincerely thanks for your valuable comments and suggestions. Hope to have the opportunity to publish on International Journal of Environmental Research and Public Health.

Looking forward to hearing from you.

Best regards,

Sincerely,

The authors

Reviewer 2 Report

Dear authors,

Let me congratulate you for the well designed and clearly described research.

Just a few minor comments and considerations.

- In the abstract (line 25), please remove good from the following statement: “The adaptation and validation of the Chinese version of the Diabetic Foot Ulcers Scale-Short Form are good", the word reliable is enough for that conclusion.

- Introduction and background are relevant to introduce the results.

- Methods: authors should explain the method they obtained the sample size calculation.

- Results are clearly described.

Author Response

Dear reviewer:

Thank you very much for your time and attention involved in the manuscript and your encouraging comments on the merits.

we also appreciate your clear and detailed feedback and hope that the explanation has adequately addressed all of your concerns. In the remainder of this letter, we will discuss each of your comments individually along with our corresponding responses.

To facilitate this discussion, we will first retype your comments in italics front and then present our responses to those comments.

Point 1:  In the abstract (line 25), please remove good from the following statement: “The adaptation and validation of the Chinese version of the Diabetic Foot Ulcers Scale-Short Form are good", the word reliable is enough for that conclusion.

Response:

Thanks for your kind suggestion. We have revised the word “good” to “reliable”. The sentence becomes: “The adaptation and validation of the Chinese version of the Diabetic Foot Ulcers Scale-Short Form are reliable and it will be a reliable instrument to evaluate the QoL of Chinese diabetic foot ulcer patients.” (Page 1, Line 36-37)

Point 2: Methods: authors should explain the method they obtained the sample size calculation.

Response:

Thanks for your detailed review. Here are the details of the sample calculation, and we have made clearer in the manuscript. The changes are as follows:

“Confirmatory Factor Analysis (CFA)[14] was conducted to test the internal consistency[15]. As the CFA is performed based on Structural equation modelling (SEM)[16], sampling followed the SEM sampling calculation method, and the sample size should be at least 200[17]. Also, it was suggested that the sample of CFA should be 100 to 200[10]. Based on all of these, we decided that the sampling size should be at least 200. In this respect, the sample size in our study was considered efficient. ”

10. Lee YN. Translation and validation of the Korean version of the Diabetic Foot Ulcer Scale-Short Form. Int Wound J. 2019;16 Suppl 1(Suppl 1):3-12. doi:10.1111/iwj.13025.

14.Arbuckle JL. IBM SPSS Amos 24 User's Guide. NY, USA: IBM: New York; 2015.) was conducted to test

15. Hung TY, Liao HC, Wang YH. Development and Validation of a Chinese Version of a School-to-Work Transition Anxiety Scale for Healthcare Students. Int J Environ Res Public Health. 2021;18(14). doi:10.3390/ijerph18147658.)

16. Cromhout A, Schutte L, Wissing MP, Schutte WD. Further Investigation of the Dimensionality of the Questionnaire for Eudaimonic Well-Being. Front Psychol. 2022;13:795770. doi:10.3389/fpsyg.2022.795770), 17. Kline RB. Principle and practice of structural equation modeling. Guilford Press; 2016.)” (Page 3, Line 104-109)

Sincerely thanks for your valuable comments and suggestions. Hope to have the opportunity to publish on International Journal of Environmental Research and Public Health.

Looking forward to hearing from you.

Best regards,

Sincerely,

The authors

Reviewer 3 Report

The goal of this study was to translate the DFS-SF into Chinese and assess its validity and reliability. The study is intriguing because Mandarin is the official language of China, which has a large population.

Recommendations

Add the full names of the abbreviations I-CVI and S-CVI/Ave CFI, RMSEA SF-36 to the abstract.

Explain why you chose Mandarin as the Chinese language's representative. Is it China's only language?

Line 65-66 The sentence is ambiguous. In addition to the five translators, are the nurse and the English professor counted as sources in translation? To clarify, the sentence should be rewritten.

67: The sentence "The primary Chinese version of the scale is as follows" appears to be meaningless in this context.

Explain the reasoning behind the sample size calculation.

Line 108: The average scale-level content validity index is abbreviated as (S-CVI), and it is mentioned in the abstract as S-CVI/Ave. unify the abbreviations

Specify the study participants' response rate. How many patients were approached, and how many were included and excluded?

Table 1: the marital status percentage is not 100; determine the correct figures.

The discussion section should be improved and supplemented with comparisons of other similar study findings.

Reference number 1: update the reference author information.

Author Response

Dear reviewer:

Thank you very much for your time and attention involved in the manuscript and your encouraging comments on the merits.

we also appreciate your clear and detailed feedback and hope that the explanation has adequately addressed all of your concerns. In the remainder of this letter, we will discuss each of your comments individually along with our corresponding responses.

To facilitate this discussion, we will first retype your comments in italics front and then present our responses to those comments.

Point 1: Add the full names of the abbreviations I-CVI and S-CVI/Ave CFI, RMSEA SF-36 to the abstract.

Response:

Thanks for your detailed review. We have added the full names of I-CVI, SCVI/Ave, CFI, and SF-36 in the abstract. And the changes are as follows: “36-Item Short-Form”. The detailed changes as follows: “The item-level content validity index (I-CVI) of the Chinese version of the Diabetic Foot Ulcer Scale varied from 0.800 to 1.000, the average scale-level content validity index (S-CVI/Ave) was 0.911, and the Cronbach’s α coefficient of the scale was 0.952. Confirmatory factor analysis indicated the good structural validity of the scale, with a Comparative Fit Index (CFI) =0.920 and root mean square error of approximation (RMSEA) of 0.069 (P <0.001). The criterion-related validity results indicated that the subscales were significantly related to the subscales of the 36-Item Short-Form Health Survey (SF-36), with coefficients ranging from 0.116 to 0.571 (P <0.05).” (Page 1, Line 23-30)

Point 2: Explain why you chose Mandarin as the Chinese language's representative. Is it China's only language?

Response:

Thanks for your detailed review. The reason why we chose Mandarin Chinese is truly because it is the official language widely used in our daily life. So, So, the Mandarin Chinese version of scale could be wodely used in regular.

Point 3: Line 65-66 The sentence is ambiguous. In addition to the five translators, are the nurse and the English professor counted as sources in translation? To clarify, the sentence should be rewritten.

Response:

Thanks for your detailed suggestion. We feel so sorry for not clarifying this part. We have rewritten the sentences. And the specific changes are as follows: “Then, we translated it into Mandarin Chinese using Brislin’s model [12] of forward and backward translation by five translators: First, a nursing graduate who did well in English and an English language professor with no medical background translated the scale into Chinese, forming the primary Chinese version of the scale was formed. Then, after an expert in diabetic foot ulcers evaluated the content equivalence, we produced the second Chinese version of the DFS-SF scale. To ensure the semantic equivalence of the Mandarin Chinese translation, a doctor unfamiliar with the original questionnaire with a bilingual background and an English-Chinese translation expert independently translated the second Chinese version back to an English version.” (Page 2, Line 76-84)

Point 4: 67: The sentence "The primary Chinese version of the scale is as follows" appears to be meaningless in this context.

Response:

Thanks for your kind suggestion. We have deleted the sentence from our manuscript.

Point 5: Explain the reasoning behind the sample size calculation.

Response:

Thanks for your detailed review. Here are the details of the sample calculation, and we make this part clearer in the manuscript. The changes are as follows:

“Confirmatory Factor Analysis (CFA)[14] was conducted to test the internal consistency[15]. As the CFA is performed based on Structural equation modelling (SEM)[16], sampling followed the SEM sampling calculation method, and the sample size should be at least 200[17]. Also, it was suggested that the sample of CFA should be 100 to 200[10]. Based on all of these, we decided that the sampling size should be at least 200. In this respect, the sample size in our study was considered efficient. ”

10. Lee YN. Translation and validation of the Korean version of the Diabetic Foot Ulcer Scale-Short Form. Int Wound J. 2019;16 Suppl 1(Suppl 1):3-12. doi:10.1111/iwj.13025.

14. Arbuckle JL. IBM SPSS Amos 24 User's Guide. NY, USA: IBM: New York; 2015.) was conducted to test

15. Hung TY, Liao HC, Wang YH. Development and Validation of a Chinese Version of a School-to-Work Transition Anxiety Scale for Healthcare Students. Int J Environ Res Public Health. 2021;18(14). doi:10.3390/ijerph18147658.)

16. Cromhout A, Schutte L, Wissing MP, Schutte WD. Further Investigation of the Dimensionality of the Questionnaire for Eudaimonic Well-Being. Front Psychol. 2022;13:795770. doi:10.3389/fpsyg.2022.795770), 17. Kline RB. Principle and practice of structural equation modeling. Guilford Press; 2016.)” (Page 3, Line 104-109)

Point 6: Line 108: The average scale-level content validity index is abbreviated as (S-CVI), and it is mentioned in the abstract as S-CVI/Ave. unify the abbreviations

Response:

Thanks for your detailed review. We feel so sorry for our carelessness. S-CVI is the abbreviation of the Scale-level content validation index. And the average scale-level content validity index is abbreviated as S-CVI/Ave.

Point 7: Specify the study participants' response rate. How many patients were approached, and how many were included and excluded?

Response:

Thank you for your detailed review. We have added the participants’ response rate, and the details are as follows: “We recruited a total of 246 patients, 23 of which rejected inclusion, for a response rate of 90.7%, and 15 participants did not finish the entire assessment. Finally, 208 patients who finished the assessment were included in our study.” (Page 4, Line 155-157)

Point 8: Table 1: the marital status percentage is not 100; determine the correct figures.

Response:

Thanks for your detailed review. We feel so sorry for our carelessness. There is a mistake in our calculation, and we have changed the percentage to 3.9. (Page 4, Table 1, from Line 161)

Point 9: The discussion section should be improved and supplemented with comparisons of other similar study findings.

Response:

Thanks for your detailed review. we have improved the discussion part and compared it with other similar studies. The details are highlighted in the manuscript.

Point 10: Reference number 1: update the reference author information.

Response:

Thanks for your detailed review. We have updated the author information of the reference number1, the details are as follows:

  1. International Diabetes Foundation. IDF Diabetes Altlas 10th Edition: International Diabetes Federation. International Diabetes Federation. 2021. https://diabetesatlas.org/.

Sincerely thanks for your valuable comments and suggestions. Hope to have the opportunity to publish on International Journal of Environmental Research and Public Health.

Looking forward to hearing from you.

Best regards,

Sincerely,

The authors

Round 2

Reviewer 3 Report

The raised comments were addressed adequately.